# Multidrug resistance pattern of *Acinetobacter* species isolated from clinical specimens referred to the Ethiopian Public Health Institute: 2014 to 2018 trend anaylsis

**Zeleke Ayenew**[1]*, **Eyasu Tigabu**[1,2], **Elias Syoum**[1], **Semira Ebrahim**[1], **Dawit Assefa**[1], **Estifanos Tsige**[1]

**1** National clinical bacteriology and mycology reference laboratory, Ethiopian Public Health Institute, Addis Ababa, Ethiopia, **2** The Global One Health Initiative, The Ohio State University, Columbus, OH, United States of America

* zelekeayenew377@gmail.com

**Data Availability Statement:** All relevant data are within the paper and its Supporting Information files.

## Abstract

### Background

*Acinetobacter* species have been a leading cause of nosocomial infections, causing significant morbidity and mortality over the entire world including Ethiopia. The most important features of *A. baumannii* are its ability to persist in the hospital environment and rapidly develop resistance to a wide variety of antibiotics. This study aimed to determine trend of antimicrobial resistance in *Acinetobacter* species over a five years period.

### Method

A retrospective data regarding occurrence and antimicrobial resistance of *Acinetobacter* species recovered from clinical specimens referred to the national reference laboratory was extracted from microbiology laboratory data source covering a time range from 2014 to 2018. Socio-demographic characteristics and laboratory record data was analyzed using SPSS 20.

### Results

A total of 102 strains of *Acinetobacter* species were analyzed from various clinical specimens. Majority of them were from pus (33.3%) followed by blood (23.5%), urine (15.6%) and body fluid (11.7%). Significant ascending trends of antimicrobial resistance was shown for meropenem (12.5% to 60.7%), ceftazidime (82.1% to 100%), ciprofloxacin (59.4% to 74.4%), ceftriaxone (87.1% to 98.6%), cefepime (80.0% to 93.3%) and pipracillin- tazobactam (67.8% to 96.3%). However, there was descending trend of antimicrobial resistance for tobramycin (56.5% to 42.8%), amikacin (42.1% to 31.4%) and trimethoprim-sulfamethoxazole (79.0 to 68.2%). The overall rate of carbapenem non-susceptible and multidrug resistance rates in *Acinetobacter* species were 56.7% and 71.6%.respectively.

**Funding:** The authors received no specific funding for this work.

**Competing interests:** The authors have declared that no competing interests exist.

## Conclusion

A five year antimicrobial resistance trend analysis of *Acinetobacter* species showed increasing MDR and resistance to high potent antimicrobial agents posing therapeutic challenge in our Hospitals and health care settings. Continuous surveillance and appropriate infection prevention and control strategies need to be strengthened to circumvent the spread of multidrug resistant pathogens in health care facilities.

## Introduction

*Acinetobacter* species are aerobic gram-negative bacilli that can cause healthcare-associated infections and can survive for prolonged periods in the environment and on the hands of healthcare workers [1]. *Acinetobacter* was first described in 1911 as *Micrococcus calcoaceticus* by Beijerinck, a Dutch microbiologist who isolated the organism from soil. Since then, it has had several names, nowadays known as *Acinetobacter* since the 1950s [2,3]. The genus *Acinetobacter* consists of more than 30 species, of which *A. baumannii*, and to a lesser extent genomic species 3 and 13TU, are mostly associated with clinical environment and nosocomial infections [3].

According to most recent scientific literature, *Acinetobacter species* are the second most common non-fermenting gram negative pathogens isolated from clinical specimens after *Pseudomonas aeruginosa* [2]. *Acinetobacter baumannii* has become increasingly responsible for causing hospital acquired infections (HAI), particularly in intensive care units (ICUs) [4]. It has been isolated from blood, sputum, skin, pleural fluid, and urine, usually in device associated infections [5]. The species are excellent biofilm producing bacteria, which facilitate their survival in hospital environments and are frequently found on the skin and in the respiratory and urinary tracts of hospitalized patients [6].

*Acinetobacter baumannii* is one of the most challenging pathogens among ESKAPE pathogens, standing for *Enterococcus faecium*, *Staphylococcus aureus*, *Klebsiella pneumoniae*, *A.baumannii*, *P.aeruginosa*, and *Enterobacteriaceae*, capable of "escaping" from common antibacterial treatments due to its particular antibiotic resistance [7]. The bacteria produce naturally occurring AmpC β-lactamases, as well as naturally occurring oxacillinases (OXAs) with carbapenemase activity [8].

Multi-drug resistance (MDR) in *Acinetobacter* species is defined as non-susceptible to at least 1 agent in $\geq 3$ antimicrobial categories [9]. The species are becoming increasingly resistant to nearly all routinely prescribed antimicrobial agents, including aminoglycosides, fluoroquinolones, and broad-spectrum β-lactams. The majority of strains are resistant to cephalosporin class of antimicrobials and resistance to carbapenems is increasingly reported [10]. Carbapenems which were once the mainstay therapy are no longer effective in controlling the infections caused by this organism. The foremost implication of infection with carbapenem resistant *A. baumannii* is the need to use "last-line" antibiotics such as colistin, polymyxin B, or tigecycline [11]. Sulbactam, a β-lactamase inhibitor, has good in vitro activity against *Acinetobacter species* and has been used successfully for treating carbapenem-resistant strains [12].

Risk factors for multi-drug resistant *Acinetobacter* colonization and infection include prolonged length of hospital stay, exposure to central venous catheterization, urinary catheterization, prior exposure to strong antimicrobials, greater severity of illness, surgery and receipt of invasive procedures [13].

People who have weakened immune systems, chronic lung disease, or diabetes, hospitalized patients, especially very ill patients on a ventilator, those with a prolonged hospital stay, those who have open wounds, or any person with invasive devices like urinary catheters are at greater risk for *Acinetobacter* infection. The infection can spread to susceptible persons by person-to-person contact or contact with contaminated surfaces [14].

Globally, MDR strains of *Acinetobacter* species are causing lethal hospital outbreaks often characterized by high morbidity and mortality. Due to the emergence of colistin resistance MDR-*A. baumannii* in clinical setting, WHO labeled the organism as critical pathogen [15]. A Center for Disease Control and Prevention (CDC) report, in the United States, in 2013, highlighted MDR *Acinetobacter* as a serious threat that causes $\approx$7,000 infections and $\approx$500 deaths each year. Nearly half of the strains isolated from persons with healthcare-associated infections reported to the CDC National Healthcare Safety Network in 2014 were carbapenem-nonsusceptible. Infections with carbapenem-resistant *A. baumannii* have been associated with death rates as high as 52% [16]. In Ethiopia, even though many studies had been conducted on antimicrobial resistance, there is still limited data showing MDR *Acinetobacter* infections associated morbidity and mortality at national level. Therefore, this study would provide information on the pattern of multidrug resistant *Acinetobacter* species isolated from different clinical specimens.

## Materials and methods

### Study design

A retrospective study design was followed to determine the prevalence of multidrug resistance *Acinetobacter* species and a trend analysis of antimicrobial susceptibility pattern among clinical specimens referred to the national reference laboratory of the Ethiopian Public Health Institute.

### Study period and area

The laboratory recorded data from 2014 to 2018 was analyzed from January 2019 to June 2019 in Addis Ababa. National clinical bacteriology and mycology reference laboratory under Ethiopian Public Health Institute is designed to receive microbiological specimens from St. Paulus Hospital, Aabet Hospital, Ras Desta Damtew Hospital, St. Peter Hospital, Yekatit Hospital, Minilik II Hospital, Federal Police Hospital, Alert Hospital and other health facilities of Addis Ababa city administration.

### Inclusion criteria

The study included all *Acinetobacter* species isolated from all ages and microbiological specimens referred to national reference laboratory during the study period. However, incomplete clinical information on patients and antimicrobial susceptibility testing report that did not comply Clinical and Laboratory Standards Institute guideline was excluded.

### Data extraction method

A laboratory recorded routine data in which all samples types analyzed was incorporated in the analysis. A standardized questionnaire was used to collect socio-demographic characteristics, clinical history, hospitalization and antibiotic treatment. All records of *Acinetobacter* species and antimicrobial susceptibility test was collected in a questionnaire and entered to SPSS version 20.

## Statistical analysis

Statistical analysis was performed using the SPSS version 20. Chi-square test and descriptive statistics (cross tab, frequency and proportion) were used to compare the trend of antimicrobial resistance rate, prevalence of MDR *Acinetobacter* species and resistance rates in empirically treated and untreated patients. P-value less than 0.05 were considered statistically significant.

## Ethical consideration

The study was conducted after ethical clearance was obtained from Ethiopian Public Health Institute (EPHI) scientific and ethical review committee (SERC). The IRB of EPHI has given us a waiver of consent to conduct the study.

## Data quality assurance

Retrospective check on quality record of media preparation per manufacturer instruction and laboratory Standard Operating Procedures (SOP) followed was conducted. We verified whether media, regents and antimicrobial agents have met expiration date and quality control parameters per clinical laboratory standard institute (CLSI). In addition, performance of quality control strains ATCC and sample storage system in laboratory was assessed from past records.

# Results

## Specimen source and demographical characteristics

A total of 102 *Acinetobacter* strains were isolated from various clinical specimens. Sixty percent of them were from males and 40% of them were from females with a mean age of 30.79 (SD ±19.18). Pus/wound was the major source of the isolates (33.3%), followed by blood (23.5%), urine (15.6%), body fluid (11.7%), ear (4.9%), cerebrospinal fluid (3.9%), tracheal aspirate (1.9%), sputum (0.9%) and throat (0.9%).

Regarding the specimen and isolate sources, the majority were from St. Paulus Hospital (30.4%) followed by Aabet Hospital (23.5%), Ras Desta Damtew Hospital (16.7%), St. Peter Hospital (9.8%), Yekatit 12 Hospital (4.9%), Minilik II Hospital (3.9%), Federal Police Hospital (2.9%), Alert Hospital (0.09%) and others from health facilities (6.8%). *Acinetobacter* species were mostly recovered from hospital acquired infection (HAI) (26.5%) followed by sepsis (20.5%) and surgical site infection (14.7%). The distribution of *Acinetobacter* species according to different clinical diagnosis of patients is indicated in Fig 1.

## Empirical treatment practice

Among the total of 102 patients, 26.5% of them provided specimen without initiation of antimicrobial therapy while 73.5% of them have taken antibiotic treatment empirically. Ceftriaxone was the most frequently prescribed (15.7%) antibiotics followed by ciprofloxacin and meropenem (7.8%). In addition, 84.3% of the patients have taken combined drugs (two or more antibiotics) before providing biological specimen. Vancomycin and ceftazidime were the most combined drugs prescribed that accounted for 9.8% (10/102) as shown in Table 1.

## Trends of Antimicrobial resistance (AMR)

All the isolates were tested for antimicrobials recommended for non sacrolytic bacteria according to CLSI guidelines. The antimicrobial resistance among isolates from empirically treated patients was proportionally high compared to empirically untreated patients [Fig 2].

The trends of antimicrobial resistance in the last two years (2017 to 2018) were as follows.

## *Acinetobacter* species in different clinical conditions

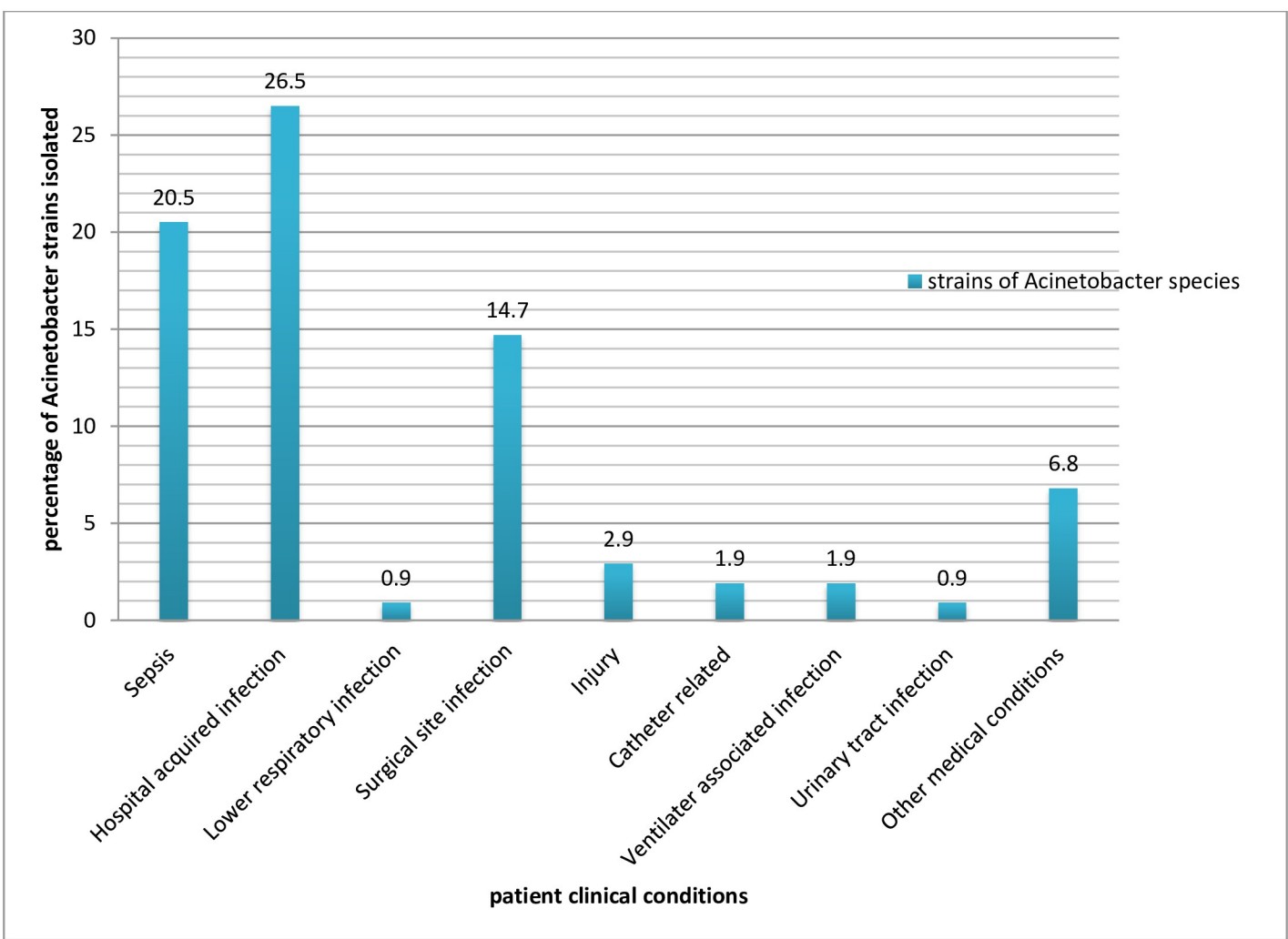

**Fig 1. Distribution of *Acinetobacter* species in relation to clinical conditions.**

**Group A antimicrobial agents.**    An increasing trend of antimicrobial resistance among antibiotics appropriate for inclusion in a routine, primary testing panel as well as routine reporting of the result for *Acinetobacter* species was observed as follows: meropenem (12.5% to 60.7%), doripenem (50.0% to 55.5%), ceftazidime (82.1% to 100%), ciprofloxacin (59.4% to 74.4%), gentamycin (55.8% to 58.0%); however, a decreasing trend of resistance was observed for tobramycin (56.5% to 42.8%) [Fig 3]. In addition, the prevalence of carbapenem non sus- ceptible *Acinetobacter* species in all tested specimen types was 56.7% (21/37).

**Group B antimicrobial agents.**    Antimicrobial resistance for second option of antibiotics have also shown ascending trend: ceftriaxone (87.1% to 88.6%), cefepime (80.0% to 93.3%), and pipracillin tazobactam (67.8% to 96.3%), Amikacin (42.1% to 31.4%), trimethoprim-sulfa- methoxazole (79.0 to 68.2%) [Fig 4].

*In vitro* activity of meropenem and doirpenem against *Acinetobacter* species were effective from 2014–2016; however, the resistance rate increased from 2017 to 2018.

**Table 1. Empirical treatment status of patients practice in health care settings from 2014–2018, Ethiopia.**

| Trends of antibiotics prescription (n = 102) | | |
|---|---|---|
| Prescription of antibiotics | Frequency | Percent |
| No antibiotics | 27 | 26.5 |
| Gentamycin | 4 | 3.9 |
| gentamycin+ceftriaxone+Vancomycin | 1 | 1.0 |
| gentamycin+cotrimoxzole+Augmentinn | 1 | 1.0 |
| Cefepime | 2 | 2.0 |
| Meropenem | 8 | 7.8 |
| Meropenem +Vancomycin | 3 | 2.9 |
| Ceftriaxone | 16 | 15.7 |
| cefepime +ceftriaxone | 1 | 1.0 |
| ceftriaxone+ciprofloxacin+Vancomycin | 4 | 3.9 |
| ceftraixone+ciprofloxacin +cotrimoxazole+cefepime | 1 | 1.0 |
| ceftriaxone +Vancomycin | 3 | 2.9 |
| ceftraiaxone+Vancomycin +ceftazidime | 1 | 1.0 |
| Ciprofloxacin | 8 | 7.8 |
| Vancomycin | 5 | 4.9 |
| Cftazidime +Vancomycin | 10 | 9.8 |
| Augmentin | 2 | 2.0 |
| OTHERS | 5 | 4.9 |
| Total | 102 | 100.0 |

The overall trends of antimicrobial classes over period of time are shown in Table 2.

## Discussions

Infection due to *Acinetobacter* species is a major challenge within the health care facilities and the community in general due to their high drug resistance even to the high potent drugs such as carbapenems. In our study 102 *Acinetobacter* species strains were analyzed to investigate the trends of antimicrobial resistance. In the analysis we found that more than 70% of patients had taken antibiotics empirically before getting confirmed culture and antimicrobial susceptibility testing (AST) result showing *Acinetobacter* infection. This problem is reported in many published works elsewhere and this could contribute to the rise of exposure to multidrug resistance infection [17,18].

The observed increase in the prevalence of MDR *Acinetobacter* was statically significant over a period of time (chi- square, = 15.8, p value = 0.003). This might be due to the fact that the laboratory capacity to detect and conduct antimicrobial resistance testing has increased over the past years.

The overall prevalence of MDR among the isolates was 71.6% which is comparable with a report from Saudi Arabia (74%) [19] and Bosnia and Herzegovina (78.4%) [20]. However, the current finding was higher than the study conducted in Jimma which isolated the MDR *Acinetobacter* species from wound specimen at a rate of 57.2% [21], and in Iran at a rate of 56.7% [22]. The antimicrobial resistance profile and prevalence of hospital associated pathogens varies from place to place.

The prevalence of MDR *Acinetobacter* from the current study is found to be lower than a previous study done in Selected Referral Hospitals in Ethiopia involving surgical site of infection (95.7%) [23] This could be due the increasing trend of early awareness of clinicians on the treatment of *Acinetobacter* infection with appropriate drugs based on the microbiology

## Antimicrobial resistance in empirical treatment

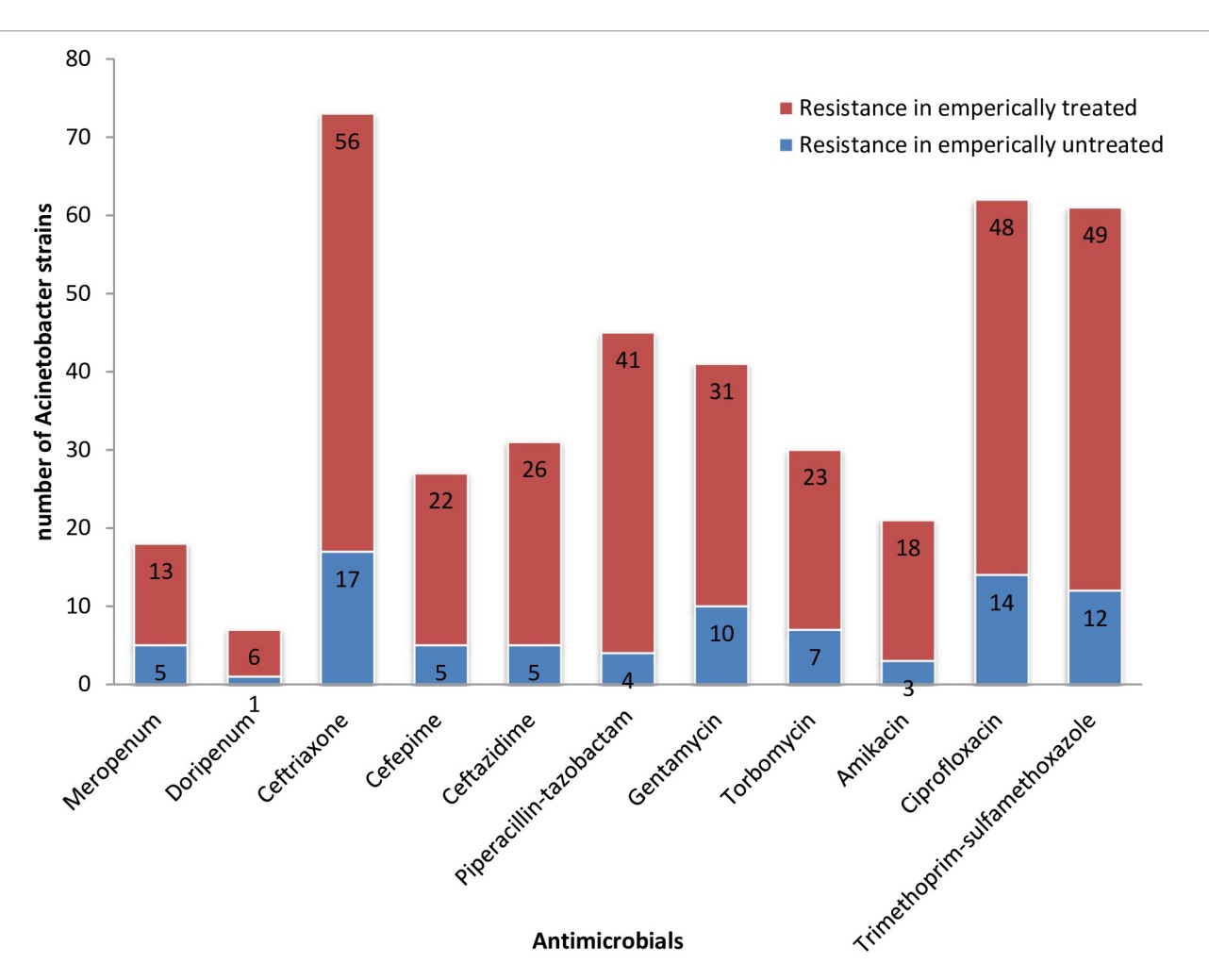

**Fig 2. Antimicrobial resistance among strains isolated from empirically treated and untreated patients referred from health facilities to Ethiopian public health institute, 2014–2018.**

laboratory identification and antimicrobials susceptibility testing results. Currently, carbapenem resistance *Acinetobacter* is included as target pathogen in the national AMR surveillance system in Ethiopia and this might also have contributed the decrease in the MDR status of *Acinetobacter* species identified in the clinical setting.

The trend analysis in antimicrobial resistance showed that in the year 2018 there was a high resistance rate for ceftriaxone (98.6%), cefepime (93.3%), ceftazidime (100%),and ciprofloxacin (74.4%) which is comparable with similar study in neighboring country Sudan which reported resistance rates as follows: ceftriaxone (95%), cefepime (92%), ceftazidime (96%) and ciprofloxacin (91%) [24].

Compared to the previous years, carbapenem resistance in *Acinetobacter* has increased by more than 50% by 2018 and this finding is comparable with previous studies conducted in East Africa [25] and in south East Asia [26].

## Trend of antimicrobial resistance for first line drugs

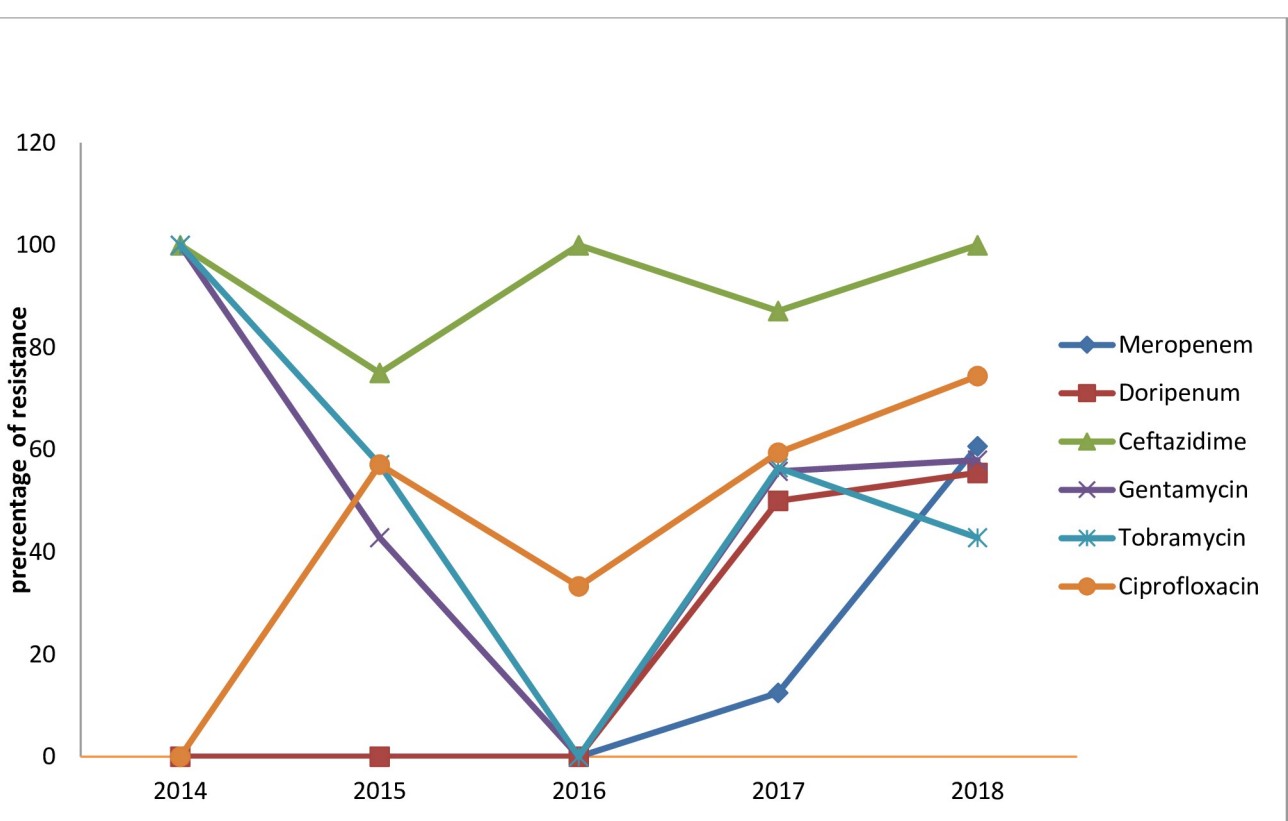

**Fig 3. Trends of antimicrobial resistance for first line drugs among the *Acinetobacter* species from 2014 to 2018.**

The resistance to aminoglycoside from the current study was found to vary among the different agents under this antimicrobial class. While resistance to gentamycin was shown to increase from 48% to 58% from 2015 to 2018, resistance to tobramycin and amikacin was below 50%. The resistance rate reported here for tobramycin is in line with a study report from Morocco [27].

## Conclusion

In conclusion, there has been an increasing trend of antimicrobials resistance in *Acinetobacter* species isolated from different sample source. *Acinetobacter* infection would remain a therapeutic challenge in our Hospitals and health care settings due to the increasing rate of *Acinetobacter* species with traits of MDR and resistance to high potent antimicrobial agents. Continuous surveillance and appropriate infection prevention and control program needs to be strengthened to circumvent the spread of these pathogens in the health care facilities.

### Limitation of the study

Although Colistin (polymyxin E) is an antibiotic used as a last-resort for multidrug-resistant gram negative infections' including *Acinetobacter*, this agent was not tested in the present study. Further molecular characterization of *Acinetobacter* exploring the genes responsible for MDR was not performed. All strains were not tested against carbapenem due to lack of supplies for first three years of the study.

## Trend of antimicrobial resistance for second line drugs

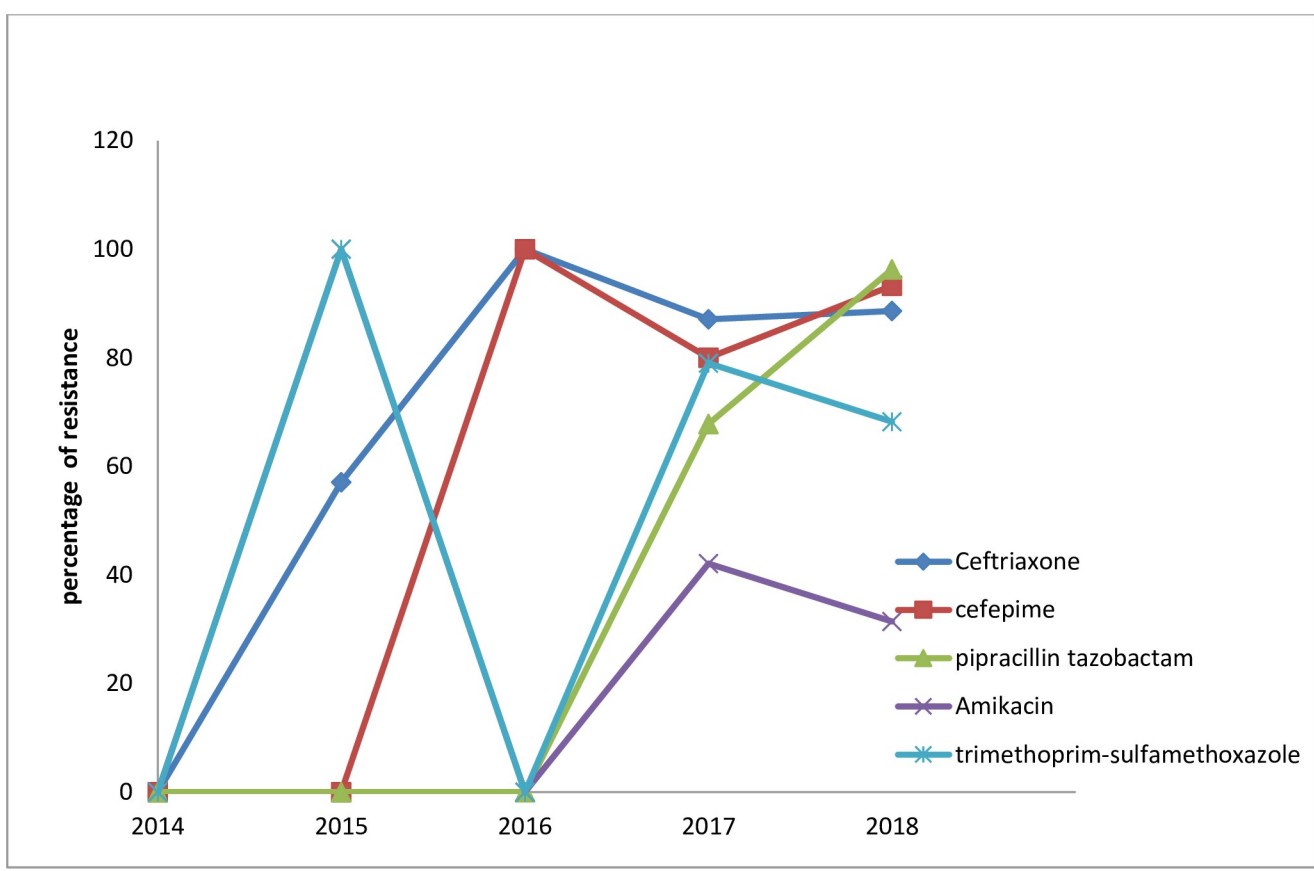

**Fig 4. Trends of antimicrobial resistance for second option of drugs among the *Acinetobacter* species from 2014 to 2018.**

**Table 2. Trend of antimicrobial resistance in *Acinetobacter* species from 2014 to 2018.**

| Antimicrobial classes | Antimicrobials | Disk content | % Resistance per year | | | | |
|---|---|---|---|---|---|---|---|
| | | | 2014 | 2015 | 2016 | 2017 | 2018 |
| Carbapenem | Meropenum | 10ug | 0 | 0 | 0 | 12.5 | 60.7 |
| | Doripenum | 10ug | 0 | 0 | 0 | 50.0 | 55.5 |
| Cephalosporin | Ceftriaxone | 30ug | 0 | 57.1 | 100 | 87.1 | 98.6 |
| | Cefepime | 30ug | 0 | 0 | 100 | 80 | 93.3 |
| | Ceftazidime | 30ug | 100 | 75 | 100 | 82.1 | 100 |
| Beta lactams | Piperacillin-tazobactam | 100/10ug | 0 | 0 | 0 | 67.8 | 96.3 |
| Aminoglycosides | Gentamycin | 10ug | 100 | 42.8 | 0 | 55.8 | 58.0 |
| | Tobramycin | 10ug | 100 | 57.1 | 0 | 56.5 | 42.8 |
| | Amikacin | 30ug | 0 | 100 | 0 | 42.1 | 31.4 |
| Fluoroquinoles | Ciprofloxacin | 5ug | 0 | 57.1 | 33.3 | 59.4 | 74.4 |
| Folate pathway inhibitors | Trimethoprim-sulfamethoxazole | 1.25/23.75ug | 0 | 100 | 0 | 79.0 | 68.2% |

## Supporting information

**S1 File.**
(ZIP)

## Acknowledgments

We express our grateful appreciation to national clinical bacteriology and mycology reference laboratory staffs Amete mihret, Negga Asamene, Rajaha Abubeker, Surafel Fentaw,Abebe Assefa, Degefu Beyene, Dejene Shiferaw, Tesfa Addis, Yonas Mekonen,Yohanise Yitagesu, Abera Abdeta, Meseret Assefa and Etsehiwot Adamu for their technical support.

## Author Contributions

**Conceptualization:** Zeleke Ayenew.

**Data curation:** Zeleke Ayenew, Semira Ebrahim, Dawit Assefa, Estifanos Tsige.

**Formal analysis:** Zeleke Ayenew, Elias Syoum, Semira Ebrahim, Dawit Assefa, Estifanos Tsige.

**Funding acquisition:** Zeleke Ayenew.

**Investigation:** Zeleke Ayenew.

**Methodology:** Zeleke Ayenew.

**Project administration:** Zeleke Ayenew.

**Resources:** Zeleke Ayenew.

**Software:** Zeleke Ayenew.

**Supervision:** Zeleke Ayenew, Eyasu Tigabu, Estifanos Tsige.

**Validation:** Zeleke Ayenew.

**Visualization:** Zeleke Ayenew, Eyasu Tigabu, Elias Syoum, Semira Ebrahim, Dawit Assefa.

**Writing – original draft:** Zeleke Ayenew.

**Writing – review & editing:** Zeleke Ayenew, Eyasu Tigabu, Elias Syoum, Semira Ebrahim.

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
