## [Decision Letter · Decision Letter 0]

5 Oct 2020

PONE-D-20-27426

Multi drug resistance pattern of Acinitobacter species isolated from clinical specimens refereed to Ethiopian public health institute, Ethiopia: a retrospective study.

PLOS ONE

Dear Dr. Ayenew,

Thank you for submitting your manuscript to PLOS ONE. After careful consideration, we feel that it has merit but does not fully meet PLOS ONE’s publication criteria as it currently stands. Therefore, we invite you to submit a revised version of the manuscript that addresses the points raised during the review process.

We look forward to receiving your revised manuscript.

Kind regards,

Monica Cartelle Gestal, PhD

Academic Editor

PLOS ONE

Journal Requirements:

2.Thank you for stating the following financial disclosure:

 [The funders had no role in study design, data collection and analysis, decision to publish, or preparation of the manuscript.].

4. Please ensure that you refer to Figure 4 in your text as, if accepted, production will need this reference to link the reader to the figure.

Reviewers' comments:

Reviewer's Responses to Questions

**Comments to the Author**

1. Is the manuscript technically sound, and do the data support the conclusions?

Reviewer #1: Yes

Reviewer #2: Partly

2. Has the statistical analysis been performed appropriately and rigorously? 

Reviewer #1: N/A

Reviewer #2: Yes

3. Have the authors made all data underlying the findings in their manuscript fully available?

Reviewer #1: Yes

Reviewer #2: Yes

4. Is the manuscript presented in an intelligible fashion and written in standard English?

Reviewer #1: No

Reviewer #2: Yes

5. Review Comments to the Author

Reviewer #1: The manuscript describes the increase of resistance rates in Acinetobacter spp. strains, isolated from different types of samples, in Ethiopia. Although the authors presented interesting results, it is difficult for me to recommend this paper for publication in PLOS ONE in its present form.

1- The authors must include line numbers, as well as page numbers to help with the correction.

2- The authors should revise the English and the style of the text. There are two types of fonts along the text.

3- The authors should correct the nomenclature, in many cases, they have written Acinitobacter instead of Acinetobacter, and on several occasions, the word appears without italics.

4- The authors should homogenize the names of the antibiotics, whether with or without a capital letter, but all the same. The same applies for the percentages, between or not a parenthesis, but all the same in the same paragraph.

5- The acronym HAI has two different explanations, one in the introduction and another one in the results.

Introduction:

6- Line 7: The authors should explain the meaning of TU.

7- The first time the name of a bacteria is mentioned should be written with its complete name. The subsequent times should be written in the reducing form of its name.

8- Line 30: b-lactamase should be changed to β-lactamase.

Material and Methods

9- The section “Study period and area” is partially missing.

10- It is not clear if “Sampling technique” is a section that includes the next ones, or if the content of this section is missing.

11- The authors should explain which antimicrobial susceptibility test and how they performed it.

12- Line 12: The authors may scape the r from strains.

13- The authors should explain what statistical test they have used.

Results

14- It would be interesting if the authors could provide data relating the resistance rates of the strains with the empirical treatment received by the patients.

15- Figure 1: The authors should include the complete name of the labels.

16- Line 9: The authors should explain the meaning of CSF.

17- Line 11: Why the authors tested carbapenem susceptibility only in 37 strains, instead of in all the 102 samples?

18- Line 11: The reference to Figure 2 should be in the previous sentence.

19- The percentage of ceftriaxone for 2018 in the text does not correspond with the 2018 percentage in the table.

20- Figure 2: The authors should explain the meaning of EPHI.

21- Figures 3 and 4: The authors should include the label of the Y-axis.

22- Figures 3 and 4: I recommend the elimination of the grey background from both figures to facilitate the differentiation between the different line colours.

23- Figures 3 and 4: the meaning of the antibiotic acronyms should be explained.

Discussion

24- Line 10: What is the meaning of x^2?

25- The authors should revise the style and the content of the discussion, especially the last three paragraphs.

Reviewer #2: English needs improvement.

Absolute numbers may not be mentioned when percentage is given.

Introduction: at risk people may be described in a single sentence.

Study period: the sentence seems incomplete.

More than half of the samples were collected from two hospitals. How was the hospitals selected for this study?

Figures should be in percentage.

The references are not uniformly written.

6. PLOS authors have the option to publish the peer review history of their article (what does this mean?). If published, this will include your full peer review and any attached files.

Reviewer #1: No

Reviewer #2: **Yes: **Manas Pratim Roy

---

## [Author Response · Author response to Decision Letter 0]

11 Nov 2020

Response to reviewers 

Journal Requirements:

Response –Thank you editor, we have edited our manuscript based on the PLOS ONE's style requirements.

 [The funders had no role in study design, data collection and analysis, decision to publish, or preparation of the manuscript.].

a. Please clarify the sources of funding (financial or material support) for your study. List the grants or organizations that supported your study, including funding received from your institution.

d. If you did not receive any funding for this study, please state: “The authors received no specific funding for this work.”

Response – thank you editor! The authors received no specific funding for this work.” We have included this in updated cover letter. 

Response –thank you we included in the method section.

4. Please ensure that you refer to Figure 4 in your text as, if accepted, production will need this reference to link the reader to the figure.

Response –thank you we referred Figure 4 in the text with some modification. 

Reviewer #1: The manuscript describes the increase of resistance rates in Acinetobacter spp. strains, isolated from different types of samples, in Ethiopia. Although the authors presented interesting results, it is difficult for me to recommend this paper for publication in PLOS ONE in its present form.

1- The authors must include line numbers, as well as page numbers to help with the correction.

Response – thank you reviewer for the suggestion to follow submission guidelines we included the line numbers and page number. 

2- The authors should revise the English and the style of the text. There are two types of fonts along the text.

Response – thank you we revised the English and make the fonts uniform ‘Arial’.

3- The authors should correct the nomenclature, in many cases, they have written Acinitobacter instead of Acinetobacter, and on several occasions, the word appears without italics.

Response –yes we admitted this and we use ‘find and replace’ Acinitobacter with Acinetobacter throughout the text. Thank you for this important comment.

4- The authors should homogenize the names of the antibiotics, whether with or without a capital letter, but all the same. The same applies for the percentages, between or not a parenthesis, but all the same in the same paragraph.

Response – thank you, we make it small letter for and antibiotics and make it uniform for percentages, between or not a parenthesis, throughout the texts as highlighted yellow. 

5- The acronym HAI has two different explanations, one in the introduction and another one in the results.

Response –thank you reviewer we make it uniform as the acronym HAI means to show hospital acquired infection. 

Introduction:

6- Line 7: The authors should explain the meaning of TU.

Response –thank you -this is a genomic species of Acinetobacter which is given by taxonomy classification in addition to species level. 

7- The first time the name of a bacteria is mentioned should be written with its complete name. The subsequent times should be written in the reducing form of its name.

Response – thank you for your comment also with exception at the beginning of a sentence, we wrote it in reduced form.

8- Line 30: b-lactamase should be changed to β-lactamase.

Material and Methods

Response –we thank you. Change has been made as β-lactamase as highlighted yellow 

9- The section “Study period and area” is partially missing.

Response – yes, it was missing. We filed it appropriately here after. Thank you! 

10- It is not clear if “Sampling technique” is a section that includes the next ones, or if the content of this section is missing.

Response – thank you again! we mean data extraction method from archived records of the laboratory.

11- The authors should explain which antimicrobial susceptibility test and how they performed it. 

Response – thank you for the comment. The test protocol for the antimicrobial susceptibility test was Kirby-Bauer disc diffusion method.

12- Line 12: The authors may scape the r from strains. 

Response- thanks you! Corrected as highlighted in the text as ‘strains’ 

13- The authors should explain what statistical test they have used.

Descriptive ,crosstab, chi-square ,ratio, proportion, 

Results

14- It would be interesting if the authors could provide data relating the resistance rates of the strains with the empirical treatment received by the patients.

Response – thank you! , we have included that in figure 2.

15- Figure 1: The authors should include the complete name of the labels.

Response- thank you we wrote the complete name

16- Line 9: The authors should explain the meaning of CSF.

Response – thank you! we mean cerebrospinal fluid as highlighted in the text yellow.

17- Line 11: Why the authors tested carbapenem susceptibility only in 37 strains, instead of in all the 102 samples?

Response –thank you! There was a shortage of the carbapenem drug during testing. 

18- Line 11: The reference to Figure 2 should be in the previous sentence.

Response – thank you! We have moved fig 2 to previous 

19- The percentage of ceftriaxone for 2018 in the text does not correspond with the 2018 percentage in the table.

Response –thank you! We mean 98.6% and that has been corrected 

20- Figure 2: The authors should explain the meaning of EPHI.

Response –thank you we mean Ethiopian public health Institute. 

21- Figures 3 and 4: The authors should include the label of the Y-axis.

Response –thank you we included the label of Y-axis.

22- Figures 3 and 4: I recommend the elimination of the grey background from both figures to facilitate the differentiation between the different line colours.

Response –thank you we eliminated the background color.

23- Figures 3 and 4: the meaning of the antibiotic acronyms should be explained.

Response –thank you, we wrote in long form.

Discussion

24- Line 10: What is the meaning of x^2?

Response –thank you x2 we mean a symbol of chi-square 

25- The authors should revise the style and the content of the discussion, especially the last three paragraphs.

Response –thank you so much we have corrected the last three paragraphs 

Reviewer #2: English needs improvement.

Response –thank you we have attempted to revise the English grammar and spelling errors.

Absolute numbers may not be mentioned when percentage is given.

Response –thank you we opted to use percentage and edit again in the abstract and result section 

Introduction: at risk people may be described in a single sentence.

Response:Thank you for the comments. Correction has been made per the comment given. 

Study period: the sentence seems incomplete.

Response .thank you, it was missed we incorporated and fill the incomplete as highlighted yellow

More than half of the samples were collected from two hospitals. How was the hospitals selected for this study?

Response –thank you, the study included all health facilities referring specimens however specimens were no growth for Acinetobacter species during culturing in some facilities. They are high load government hospitals refer sample for microbiology culture and antimicrobial susceptibility testing. This study analyzed the isolates only i.e. we excluded the no growth specimens 

Figures should be in percentage.

Response .Thank you we have made change to percentage 

The references are not uniformly written.

Response –thank you we revised and have written it in Vancouver style.

---

## [Decision Letter · Decision Letter 1]

13 Jan 2021

PONE-D-20-27426R1

Multidrug resistance pattern of Acinetobacter species isolated from clinical specimens referred to Ethiopian Public Health Institute: a retrospective study.

PLOS ONE

Dear Dr. Ayenew,

Thank you for submitting your manuscript to PLOS ONE. After careful consideration, we feel that it has merit but does not fully meet PLOS ONE’s publication criteria as it currently stands. Therefore, we invite you to submit a revised version of the manuscript that addresses the points raised during the review process.

Please pay special attention to the comments suggested by the reviewers are they still highlight major concenrs that will improve the quality and clarity of the manuscrtpt.

We look forward to receiving your revised manuscript.

Kind regards,

Monica Cartelle Gestal, PhD

Academic Editor

PLOS ONE

Reviewers' comments:

Reviewer's Responses to Questions

**Comments to the Author**

1. If the authors have adequately addressed your comments raised in a previous round of review and you feel that this manuscript is now acceptable for publication, you may indicate that here to bypass the “Comments to the Author” section, enter your conflict of interest statement in the “Confidential to Editor” section, and submit your "Accept" recommendation.

Reviewer #1: All comments have been addressed

Reviewer #3: All comments have been addressed

2. Is the manuscript technically sound, and do the data support the conclusions?

Reviewer #1: (No Response)

Reviewer #3: Partly

3. Has the statistical analysis been performed appropriately and rigorously? 

Reviewer #1: (No Response)

Reviewer #3: Yes

4. Have the authors made all data underlying the findings in their manuscript fully available?

Reviewer #1: (No Response)

Reviewer #3: Yes

5. Is the manuscript presented in an intelligible fashion and written in standard English?

Reviewer #1: (No Response)

Reviewer #3: Yes

6. Review Comments to the Author

Reviewer #1: The paper has been clearly improve, but I would recommend the authors to revise again due to some italics, antibiotic names as well as some spaces that are missing.

Reviewer #3: the manuscript needs some attention on the following -

1. Introduction- please clearly define the multi-drug resistance of Acinetobacter species.

globally and regional data supporting the severity of multi-drug resistance of the organism (eg-mortality or

morbidity) would make the context stronger.

2. Methods- Inclusion criteria are not mentioned clearly.

3. Discussion- The result of empirical therapy is not discussed. please discuss with significance.

4. Conclusion- Please rephrase the 1st line of conclusion.

7. PLOS authors have the option to publish the peer review history of their article (what does this mean?). If published, this will include your full peer review and any attached files.

Reviewer #1: No

Reviewer #3: **Yes: **Iffat Ara Ifa

---

## [Author Response · Author response to Decision Letter 1]

29 Jan 2021

Review Comments to the Author

Reviewer #1: The paper has been clearly improved, but I would recommend the authors to revise again due to some italics, antibiotic names as well as some spaces that are missing.

Response – thank you! We have made changes in our revision some italics, antibiotic names as well as some spaces missed.

Reviewer #3: the manuscript needs some attention on the following -

1. Introduction- please clearly define the multi-drug resistance of Acinetobacter species.

globally and regional data supporting the severity of multi-drug resistance of the organism (eg-mortality or

morbidity) would make the context stronger.

Response – thank you for the comments to be added and we incorporate definition of MDR –Acinetobacter species, global and regional data supporting the severity of the disease in the introduction part.

2. Methods- Inclusion criteria are not mentioned clearly.

Response – thank you we have added the inclusion criteria of the study in method part 

3. Discussion- The result of empirical therapy is not discussed. please discuss with significance.

Response – thank you for the comment regarding missed discussion of empirical therapy. We have added with significance in line number 205 on page 11 

4. Conclusion- Please rephrase the 1st line of conclusion.

Response – we rephrase the 1st line of conclusion in line 243 on page 13 of the manuscript.

---

## [Decision Letter · Decision Letter 2]

16 Apr 2021

Multidrug resistance pattern of Acinetobacter species isolated from clinical specimens referred to the Ethiopian Public Health Institute: 2014 to 2018 trend anaylsis

PONE-D-20-27426R2

Dear Dr. Ayenew,

We’re pleased to inform you that your manuscript has been judged scientifically suitable for publication and will be formally accepted for publication once it meets all outstanding technical requirements.

Kind regards,

Monica Cartelle Gestal, PhD

Academic Editor

PLOS ONE

Additional Editor Comments (optional):

Reviewers' comments:

Reviewer's Responses to Questions

**Comments to the Author**

1. If the authors have adequately addressed your comments raised in a previous round of review and you feel that this manuscript is now acceptable for publication, you may indicate that here to bypass the “Comments to the Author” section, enter your conflict of interest statement in the “Confidential to Editor” section, and submit your "Accept" recommendation.

Reviewer #1: All comments have been addressed

Reviewer #3: All comments have been addressed

2. Is the manuscript technically sound, and do the data support the conclusions?

Reviewer #1: (No Response)

Reviewer #3: Yes

3. Has the statistical analysis been performed appropriately and rigorously? 

Reviewer #1: (No Response)

Reviewer #3: Yes

4. Have the authors made all data underlying the findings in their manuscript fully available?

Reviewer #1: (No Response)

Reviewer #3: Yes

5. Is the manuscript presented in an intelligible fashion and written in standard English?

Reviewer #1: (No Response)

Reviewer #3: Yes

6. Review Comments to the Author

Reviewer #1: (No Response)

Reviewer #3: Thanks to the author for the corrections properly.

I have no other issues regarding this article. If there is any minor issue please correct it.

7. PLOS authors have the option to publish the peer review history of their article (what does this mean?). If published, this will include your full peer review and any attached files.

Reviewer #1: No

Reviewer #3: **Yes: **Iffat Ara Ifa

---

## [Editor Report · Acceptance letter]

21 Apr 2021

PONE-D-20-27426R2 

Multidrug resistance pattern of *Acinetobacter* species isolated from clinical specimens referred to the Ethiopian Public Health Institute: 2014 to 2018 trend anaylsis     

Dear Dr. Ayenew:

I'm pleased to inform you that your manuscript has been deemed suitable for publication in PLOS ONE. Congratulations! Your manuscript is now with our production department. 

Kind regards, 

on behalf of

Dr. Monica Cartelle Gestal 

Academic Editor

PLOS ONE